# Cu-mediated enantioselective C–H alkynylation of ferrocenes with chiral BINOL ligands

Xin Kuang[1,2], Jian-Jun Li[1], Tao Liu[1], Chang-Hua Ding[2], Kevin Wu[3], Peng Wang [1,4] ✉ & Jin-Quan Yu [3] ✉

A wide range of Cu(II)-catalyzed C–H activation reactions have been realized since 2006, however, whether a C–H metalation mechanism similar to Pd(II)-catalyzed C–H activation reaction is operating remains an open question. To address this question and ultimately develop ligand accelerated Cu(II)-catalyzed C–H activation reactions, realizing the enantioselective version and investigating the mechanism is critically important. With a modified chiral BINOL ligand, we report the first example of Cu-mediated enantioselective C–H activation reaction for the construction of planar chiral ferrocenes with high yields and stereoinduction. The key to the success of this reaction is the discovery of a ligand acceleration effect with the BINOL-based diol ligand in the directed Cu-catalyzed C–H alkynylation of ferrocene derivatives bearing an oxazoline-aniline directing group. This transformation is compatible with terminal aryl and alkyl alkynes, which are incompatible with Pd-catalyzed C–H activation reactions. This finding provides an invaluable mechanistic information in determining whether Cu(II) cleaves C–H bonds via CMD pathway in analogous manner to Pd(II) catalysts.

In the past decade, enantioselective C–H activation reactions via asymmetric metalation have emerged as a promising platform for asymmetric catalysis[1–4]. Significant progress has been made in this field with the development of catalytic systems using Pd(II)[5–7], Pd(0)[8–12], Rh(III)[13–16], Rh(I)[17], Ir[18–25], and other precious metals. In particular, a wide range of chiral Pd(II) catalysts have been developed for inducing point chirality, axial chirality, and planar chirality owing to the discovery of bifunctional chiral mono-*N*-protected amino acid (MPAA) ligands and next-generation ligands capable of accelerating C–H activation[5–7]. In contrast, there are few reports of asymmetric C–H activation using first-row transition metal catalysts, such as Fe, Co, Ni, or Cu, despite their abundance and low cost[26–33]. For example, Cu-catalyzed C–H activation reactions have immense potential utility given the superior tolerance of Cu catalysts for heterocycles and other sensitive functionalities compared to precious metal catalysts[34–54]. However, the development of enantioselective transformations has not been successful. The discovery of chiral ligands that accelerate Cu-catalyzed C–H activation is essential, as it enables the chiral ligand–catalyst complex to outcompete the racemic "background" reaction that is prevalent with unligated Cu. In addition, the establishment of a stereo model will provide unique insights into the transition state structures.

Given the aforementioned advantages of Cu-catalyzed C–H activation reactions, our group has devoted extensive effort to the development of Cu catalysis since the first report in 2006[34]. We have developed diverse transformations using a wide range of coupling partners enabled by our class of practical oxazoline-aniline directing groups (DGs)[46–54]. Our oxazoline-aniline DG not only unlocks reactivity with an enormous range of coupling partners for C–H activation, including free amines/amides[46] and trifluoromethylating reagents[48] but also demonstrates exceptional tolerance for otherwise challenging functional groups, especially a broad range of heterocycles[52,53].

[1]State Key Laboratory of Organometallic Chemistry, Shanghai Institute of Organic Chemistry, CAS 345 Lingling Road, Shanghai 200032, P.R. China. [2]School of Science, Shanghai University, 99 Shang-Da Road, Shanghai 200444, P. R. China. [3]The Scripps Research Institute (TSRI), 10550 North Torrey Pines Road, La Jolla, CA 92037, USA. [4]School of Chemistry and Materials Science, Hangzhou Institute for Advanced Study, University of Chinese Academy of Sciences, 1 Sublane Xiangshan, Hangzhou 310024, P.R. China. ✉e-mail: pengwang@sioc.ac.cn; yu200@scripps.edu

**a.** Stereoselective transition metal-catalyzed C–H activation of ferrocenes

**b.** Ligand-accelerated enantioselective Cu-mediated C–H ferrocene alkynylation

up to 82% yield, up to 93:7 er

· Ligand acceleration · Mild conditions · Broad scope · Heterocycle tolerance

**Fig. 1 | Ligand accelerated Cu-mediated enantioselective C–H alkynylation of ferrocenes. a** Stereoselective transition metal-catalyzed C–H activation of ferrocenes. DG directing group, FG functional group. **b** Ligand-accelerated enantioselective Cu-mediated C–H ferrocene alkynylation.

In the realm of stereoselective Cu-catalyzed C–H activation, we recently reported the diastereoselective C–H thiolation of ferrocenes using a chiral oxazoline directing group (Fig. 1a)[55]. Despite its practical limitations, this chiral auxiliary approach represents a promising lead for the ultimate development of Cu-catalyzed enantioselective C–H activation reactions. Here, we report the discovery of BINOL-derived ligands that accelerate the Cu-mediated C–H alkynylation of ferrocenes with an oxazoline-amide directing group. When using (S)−6,6′-dibromo-BINOL as the chiral ligand, the Cu-mediated enantioselective C–H alkynylation of ferrocene carboxylic acid derivatives was achieved with high enantioselectivity (Fig. 1b). Mechanistic studies support the presence of ligand acceleration and indicate the acceleration occurs at the key C–H cleavage step of the catalytic cycle.

## Results and discussion

### Reaction optimization for Cu-mediated C–H alkynylation

We targeted the *ortho*-C($sp^2$)–H activation of planar chiral ferrocene carboxylic acid derivatives as the model system to develop ligands to accelerate C–H activation with Cu. We derivatized the carboxylic acid with our oxazoline-amide directing group as it promotes a range of transformations and is readily installed and removed[46–54]. Achieving ligand acceleration could enable the development of asymmetric Cu-catalyzed C–H activations. A number of catalytic asymmetric C–H activations of ferrocenes using other metals have been reported[18,56–65].

Planar chiral ferrocenes are prevalent in synthetic chemistry, materials science, and medicinal chemistry[66–69], as well as privileged scaffolds for chiral ligands and catalysts[70–75]. More specifically, *ortho*-substituted planar chiral ferrocene carboxylic acids and their derivatives, the products of this method, are highly valuable in the preparation of chiral monodentate (carboxylates)[76] and bidentate (oxazolines/phosphines, etc.)[77] ligands. Traditional directed *ortho* lithiation approaches for the enantioselective preparation of *ortho*-substituted ferrocene carboxylic acids rely on chiral auxiliaries[78–82]. They suffer from efficiency and atom economy losses associated with the use of chiral auxiliaries, as well as the scope limitations associated with organolithium reagents. A Cu-catalyzed enantioselective C–H activation approach towards *ortho*-substituted planar ferrocene carboxylic acid derivatives would circumvent these limitations and permit direct installation of diverse functionality using an abundant base metal catalyst. Notably, very few enantioselective catalytic C–H activation reactions to prepare *ortho*-substituted ferrocene carboxylic acid derivatives are known[83].

Preliminary experiments with the racemic Cu-mediated C–H alkynylation of ferrocene carboxylic acid substrate **1** with

ethynylbenzene (**2a**) provided only 14% yield of product **3a** under our previously developed conditions for Cu-mediated C–H alkynylation, which did not use an exogenous ligand[36]. Through a systematic evaluation of reaction parameters, the yield of **3a** was improved to 56% (17/1.0 *mono/di*). Encouraged by our success in developing ligands capable of accelerating Pd-catalyzed C–H activation, we then initiated a search for ligand acceleration effects for this Cu-mediated transformation. A range of ligand scaffolds were systematically investigated (Fig. 2, see Supplementary Information for more details). Common mono- and bi-dentate donating ligands such as triphenylphosphine (**L1**), BINAP (**L2**), 1,10-phenanthroline (**L3**), and bis-oxazoline (**L4**) either inhibited or showed no beneficial impact on reactivity. We next evaluated mono-protected amino acid (**L5**) and pyridone (**L6**) ligands, two scaffolds widely used in Pd-catalyzed C–H activation and capable of directly accelerating the key C–H activation step. Unfortunately, both ligand classes were ineffective in this transformation. Pyridine-type (**L7**) ligands, which are also widely used to improve the reactivity of Pd C–H activation catalysts, failed to show any improvement. Finally, mono-oxazoline ligands (**L8**), which are effective ligands in Cu-mediated C–H activations with monodentate directing groups[43], were not beneficial in this transformation.

In contrast to the results with established ligands for Pd-catalyzed C–H activations, we found that (S)-BINOL (**L10**) showed a significant improvement in reactivity, increasing the yield of **3a** to 69% and indicating the potential for ligand accelerated catalysis[84]. Other bisphenol scaffolds such as **L9** and (S)-spirosilabiindane diol (SPSiOL, **L12**)[85] showed inferior results, while (S)-spirobiindane diol (SPINOL, **L11**) demonstrated similar reactivity as **L10**. Given the established modularity and ease of derivatization of the BINOL core, we chose the BINOL platform for further investigation. We first evaluated 3,3′-Me₂-BINOL (**L15**), which to our delight gave an improved yield of 76% (4.8/1 *mono/di*). Control experiments revealed that both phenolic hydroxyl groups are crucial for the high reactivity (**L13**, **L14** vs. **L10**). The observation of this potential ligand acceleration effect with Cu not only will promote the discovery of Cu-mediated/catalyzed C–H activations with this abundant and economical metal but also enable the development of enantioselective catalysis (vide infra).

### Substrate scope for the Cu-mediated C–H alkynylation

With the optimal conditions in hand, the scope of the ligand-accelerated Cu-mediated C–H alkynylation was investigated. Employing ferrocene monocarboxylic acid-derived **1** as the model substrate, we first evaluated the breadth of terminal alkynes (Fig. 3). In general, this method could tolerate a series of aryl, alkenyl, and alkyl-

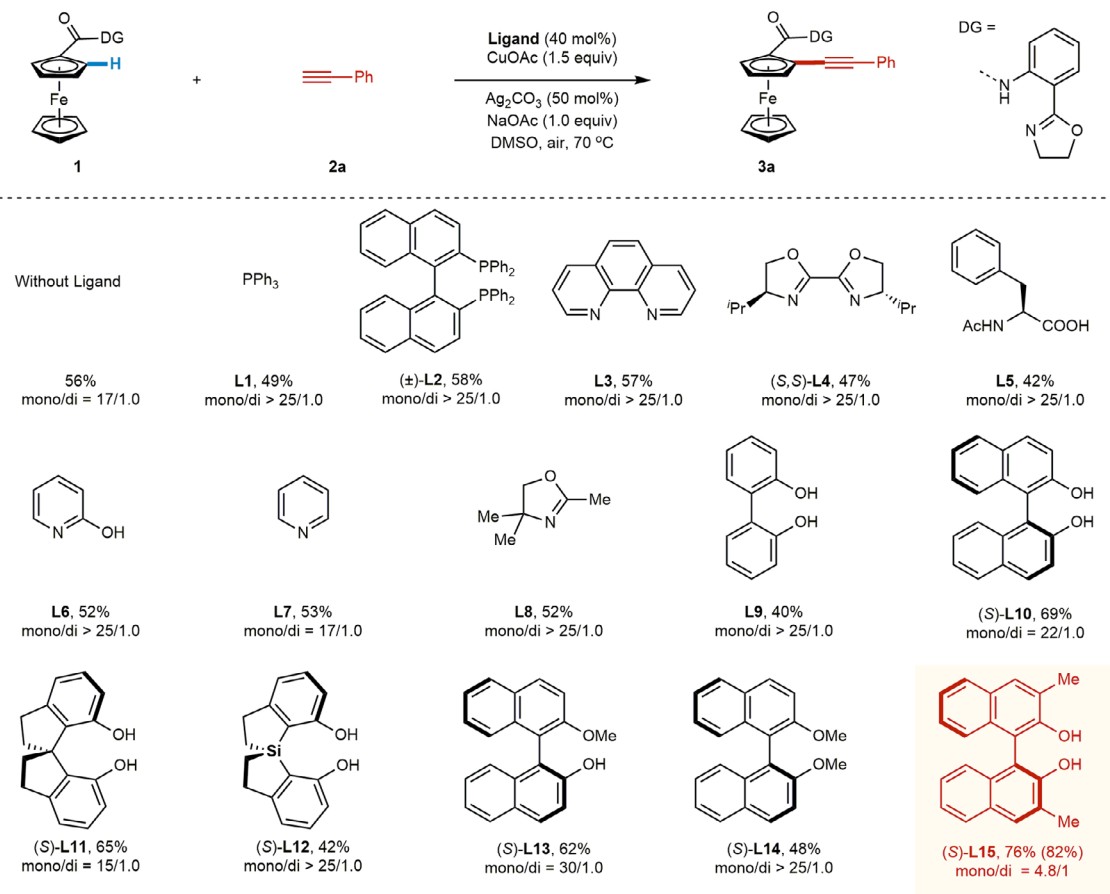

**Fig. 2 | Ligand effects for Cu-mediated C−H alkynylation of ferrocenes.** Reaction conditions: **1** (37.4 mg, 0.1 mmol, 1.0 equiv), **2a** (27.5 μL, 0.25 mmol), CuOAc (18.4 mg, 0.15 mmol), **Ligand** (40 mol%), NaOAc (8.2 mg, 0.1 mmol), DMSO (5.0 mL), 70 °C, air, 12 h. The yield was determined using tetrachloroethane as the internal standard, and the ratio of mono/di was determined by analysis of the crude $^1$H NMR. The data in the bracket indicates the isolated yield.

substituted terminal alkynes, providing the desired products in moderate to good yields (**3a**−**r**). Both electron-rich (**2b**−**d**, **2g**, **2i**) and electron-deficient (**2e**, **2f**, **2h**, **2j**) substituents on the phenyl group of phenylacetylenes are compatible with this protocol. To our delight, terminal alkyne **2l** bearing a heterocyclic thiophene moiety was also compatible, providing product **3l** in 64% yield. Notably, aliphatic alkynes containing both cyclic (**2m**, **2n**) and acyclic (**2o**−**q**) alkyl groups, were all compatible, giving products **3m**−**q** in good yields. Conjugated enyne **2r** gave the ferrocene-containing enyne **3r** in 62% yield, further demonstrating the generality of this protocol.

Next, the scope of ferrocene substrates was evaluated with phenylacetylene (**2a**) as the alkynylating reagent (**5a**−**i**), as presented in Fig. 4. Ferrocenes bearing both electron-deficient ketone and ester groups (**4a**−**e**) and electron-rich alkyl (**4f**−**h**) substituents on the lower cyclopentadiene ring were all tolerated, affording the corresponding alkynylated ferrocene derivatives in moderate to good yields. Bulky ferrocene **4i** bearing a Cp* on the lower ring gave a lower yield (38%) with excellent mono selectivity, potentially due to steric hindrance.

### Reaction optimization for Cu-mediated asymmetric C−H alkynylation

Encouraged by the potential ligand acceleration effect in Cu-mediated C−H activation with BINOL ligands, we hypothesized that an asymmetric Cu-mediated C−H alkynylation of ferrocenes could be achieved with enantioenriched chiral diol ligands. This method would enable the enantioselective preparation of useful planar chiral ferrocene scaffolds. Ligand acceleration of the catalytic cycle is particularly critical for asymmetric Cu C−H activation catalysis, in light of the prominent

achiral "background" reaction often observed with Cu. Upon evaluating enantiopure BINOL ligands in our racemic reaction protocol, we obtained a promising initial hit with (S)-BINOL ((S)-**L10**) providing product **3a** in 78.5:21.5 er. Additional optimization of reaction parameters to a mixture of CuOAc (50 mol%) and Cu(OAc)$_2$ (70 mol%) as the Cu sources and 60 mol% of (S)-**L10** ligand, as well as the use of Ag$_2$CO$_3$ (0.5 equiv) and NaOAc (1.0 equiv) improved the enantioselectivity to 82:18 er in 63% yield (6.0/1.0 mono/di) (Fig. 5). Next, ligand effects were explored. (S)-H$_8$-BINOL (**L16**) led to 54:46 er, while (S)-SPINOL (**L11**) and (R,R)-TADDOL (**L17**) gave the racemic product. Although (S)-3,3'-Me$_2$-BINOL (**L15**) showed superior reactivity under the racemic reaction conditions, it only provided the product at 55:45 er. The installation of other substituents at the 3,3'-positions of BINOL (**L18**−**20**) generally resulted in inferior selectivity to **L10**, although results with **L19** and **L20** (70:30 er, 74:26 er, respectively) suggested that electron-withdrawing groups could benefit asymmetric induction by rendering the phenolic groups more acidic. Following this reasoning, we then evaluated (S)−6,6'-Br$_2$-BINOL [(S)-**L21**], with distal Br substituents capable of acidifying the phenolic groups without steric interference. Gratifyingly, these changes improved stereocontrol to 87:13 er. Phenyl substituents at the 6,6'-positions provided inferior results (**L22**, 72:28 er). Control experiments with enantioenriched mono-((S)-**L13**) and di-((S)-**L14**) protected BINOL ligands showed that both hydroxyl groups are essential for selectivity and reactivity.

### Substrate scope for the asymmetric C−H alkynylation

With (S)−6,6'-dibromo-BINOL [(S)-**L21**] as the optimal ligand, the enantioselectivity was further improved to 92.5:7.5 by reducing the

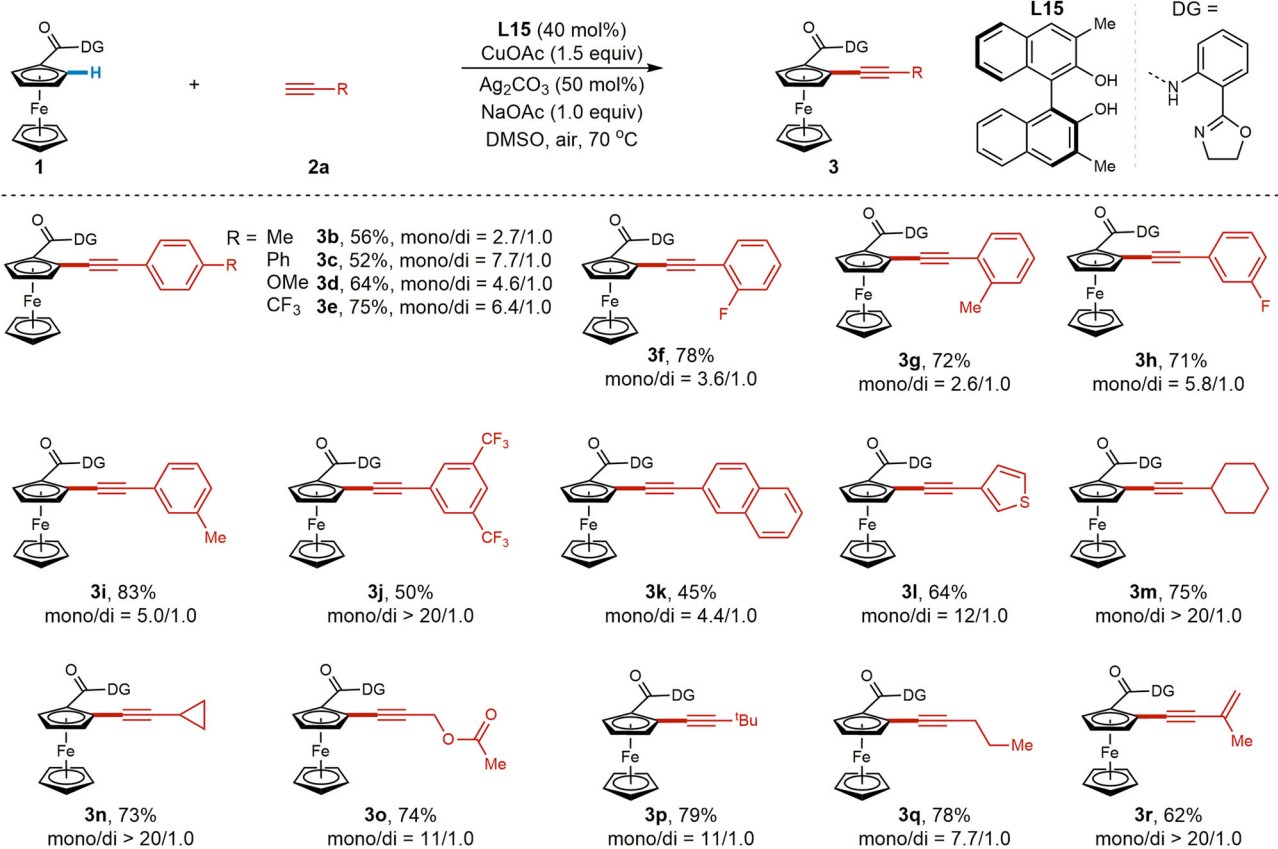

**Fig. 3 | Scope of alkynes for ligand accelerated Cu-mediated C−H alkynylation of ferrocenes.** Reaction conditions: **1** (37.4 mg, 0.1 mmol), **2** (0.25 mmol, 2.5 equiv), CuOAc (18.4 mg, 0.15 mmol), **L15** (12.6 mg, 40 mol%), NaOAc (8.2 mg, 0.1 mmol), DMSO (5.0 mL), 70 °C, air, 12 h. Isolated yield and the ratio of mono/di were determined by analysis of the crude ¹H NMR. For **3f**, the reaction was conducted for 16 h.

loading of CuOAc/[(S)-**L21**] and lowering the reaction temperature. The absolute configuration of the planar chiral alkynylated ferrocene was determined by X-ray crystallography of (Rₚ)-**3a**. This novel process features broad substrate scope with respect to the terminal alkynes (**3a–r**, Fig. 6). Both substituted phenylacetylenes and alkyl-substituted acetylenes are suitable substrates, delivering the enantioenriched mono-alkynylated ferrocenes in synthetic useful yields (35–68% yields) with good enantioselectivities (88:12 er to 93:7 er). The mono/di selectivity is lower than the racemic reaction (Figs. 3 and 4), probably due to the higher reactivity of (S)−6,6′-dibromo-BINOL [(S)-**L21**] in comparison with 3,3′-Me₂-BINOL (**L15**). Notably, heterocyclic arylalkynes containing thiophene (**2l**) also proceeded in high enantioselectivity (89.5:10.5 er). A range of ferrocene carboxylic acid derivatives (**4f–h**) are also compatible with this protocol, providing the desired planar chiral ferrocenes (Rp)-**5f-h** in high yields and high ers.

To demonstrate the synthetic utility of the Cu-mediated asymmetric C−H alkynylation, we conducted the transformation on a gram-scale and elaborated the product to synthetically useful compounds. First, the gram-scale reaction was conducted with 70 mol% of Cu(OAc)₂, 20 mol% of CuOAc, and 20 mol% of (S)-**L21** (Fig. 7a), providing enantioenriched mono-alkynylated product **3a** in 52% yield with slightly lower enantioselectivity (90:10 er), and the di-product (**di-3a**) in 7% yield. Recrystallization from hexane/dichloroethane provided **3a** in very high selectivity (99.5:0.5 er) and 38% yield. To explore potential applications of the product (Fig. 7b), we derivatized **3a** through a synthetic sequence, first through hydrogenation to *ortho*-alkylated **6a**, followed by removal of the directing group to reveal ferrocene carboxylic acid **7a**, with minimal loss of enantioenrichment (98:2 er). *Ortho*-substituted planar chiral ferrocene carboxylic acids have recently been shown by Matsunaga to be effective chiral ligands in

Co-catalyzed enantioselective C(sp³)−H activation reactions[76]. As a demonstration of our method's potential impact, we employed **7a** as a chiral ligand in the Co-catalyzed C(sp³)−H amidation of thioamide **8** (Fig. 7b), providing product **10** in high yield (95%) and promising enantioselectivity (70.5:29.5 er). This example highlights the synthetic utility of our Cu-mediated enantioselective C−H alkynylation, which enables access to novel chemical space in valuable *ortho*-substituted planar chiral ferrocenes.

## Mechanistic insights

The increased reactivity and high enantioselectivity imparted by the BINOL ligands were suggestive of ligand-accelerated catalysis, particularly in light of the strong background reaction. To further probe the possibility of ligand acceleration and collect data on the catalytic cycle, we then conducted some preliminary mechanistic studies on the system. We first performed a one-pot intermolecular competition kinetic isotope effect (KIE) experiment using **1a** and *ortho*-deuterated **D₂−1a** (Fig. 8a) in the presence of **L21**. A primary KIE was observed (4.9), indicating that C−H cleavage is likely the rate-determining step. This result indicates that the Cu-mediated C−H alkynylation is likely proceeding through CMD C−H activation at Cu, as opposed to SET pathways that do not involve Cu in C−H bond breaking, which would show no KIE (-1)[86]. This is a critical mechanistic insight as it both establishes that Cu cleaves C−H bonds in a similar manner as Pd and suggests a plausible mechanism for ligand involvement and acceleration. Having identified C−H cleavage at Cu as the rate-determining step, we next conducted initial rate studies to test for ligand acceleration (Fig. 8b). We observed a striking initial rate difference when comparing catalyst reactivity with and without BINOL-derived ligand (S)-**L21**. A rate acceleration of 11.6 times was observed with (S)-**L21**, strongly

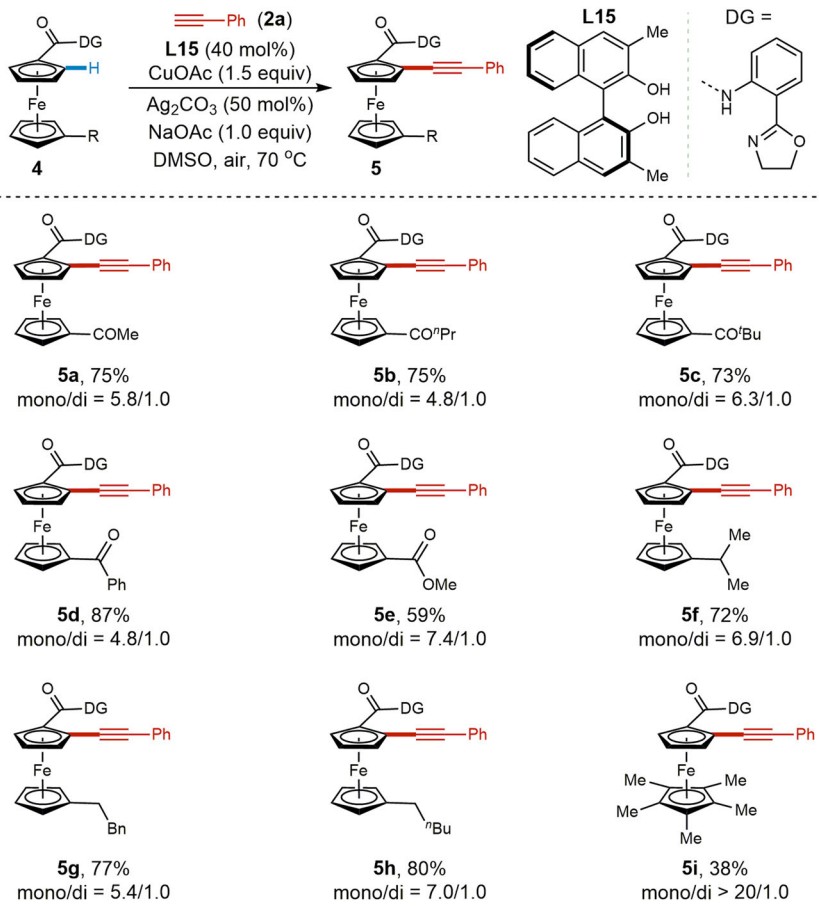

**Fig. 4 | Scope of ferrocene derivatives.** Reaction conditions: **4** (0.1 mmol, 1.0 equiv), **2a** (27.5 μL, 0.25 mmol), CuOAc (18.4 mg, 0.15 mmol), **L15** (12.6 mg, 40 mol%), NaOAc (8.2 mg, 0.1 mmol), DMSO (5.0 mL), 70 °C, air, 12 h. Isolated yield and the ratio of mono/di was calculated according to the isolated yield.

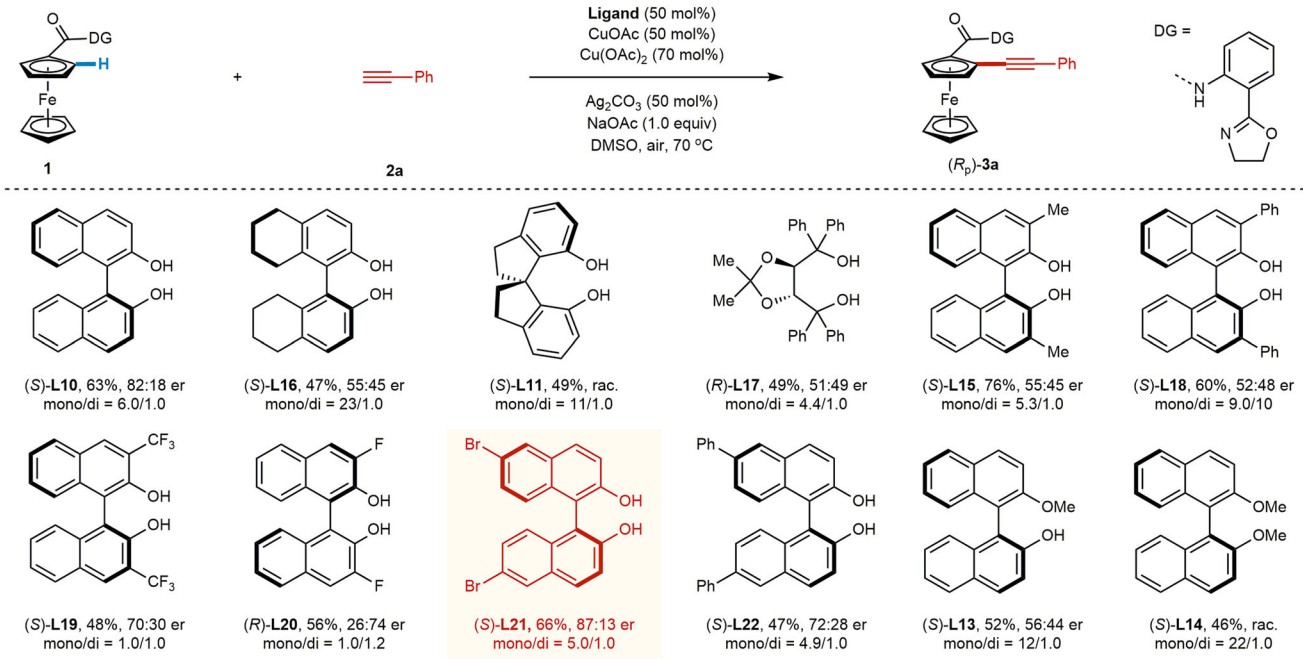

**Fig. 5 | Evaluation of chiral diol ligands for Cu-mediated asymmetric C−H alkynylation.** Reaction conditions: **1** (37.4 mg, 0.1 mmol), **2a** (33.0 μL, 0.30 mmol), CuOAc (6.1 mg, 50 mol%), chiral diol ligand (50 mol%), Cu(OAc)$_2$ (12.7 mg, 0.07 mmol), NaOAc (8.2 mg, 0.1 mmol), DMSO (5.0 mL), 60 °C, air, 12 h. Isolated yield and the ratio of mono/di were determined by analysis of the crude $^1$H NMR. The er value for the mono-alkynylated product was determined by chiral HPLC.

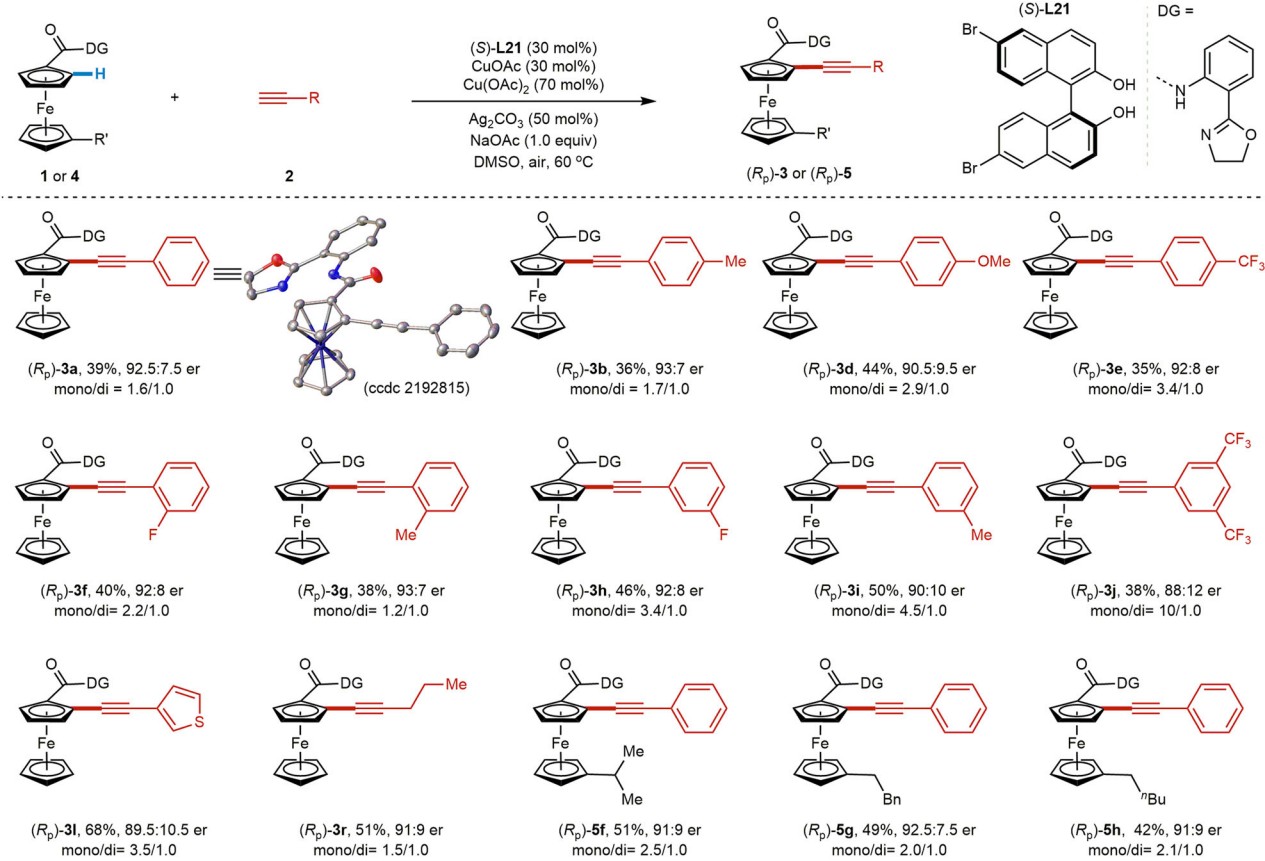

**Fig. 6 | Cu-mediated enantioselective C–H alkynylation of ferrocenes.** Reaction conditions: **1** or **4** (0.1 mmol), **2** (0.25 mmol), CuOAc (3.7 mg, 30 mol%), **(S)-L21** (13.3 mg, 30 mol%), Cu(OAc)$_2$ (12.7 mg, 0.07 mmol), NaOAc (8.2 mg, 0.1 mmol), DMSO (5.0 mL), 60 °C, air, 12 h. Isolated yield and the ratio of mono/di were determined by analysis of the crude $^1$H NMR. The er value for the mono-alkynylated product was determined by chiral HPLC.

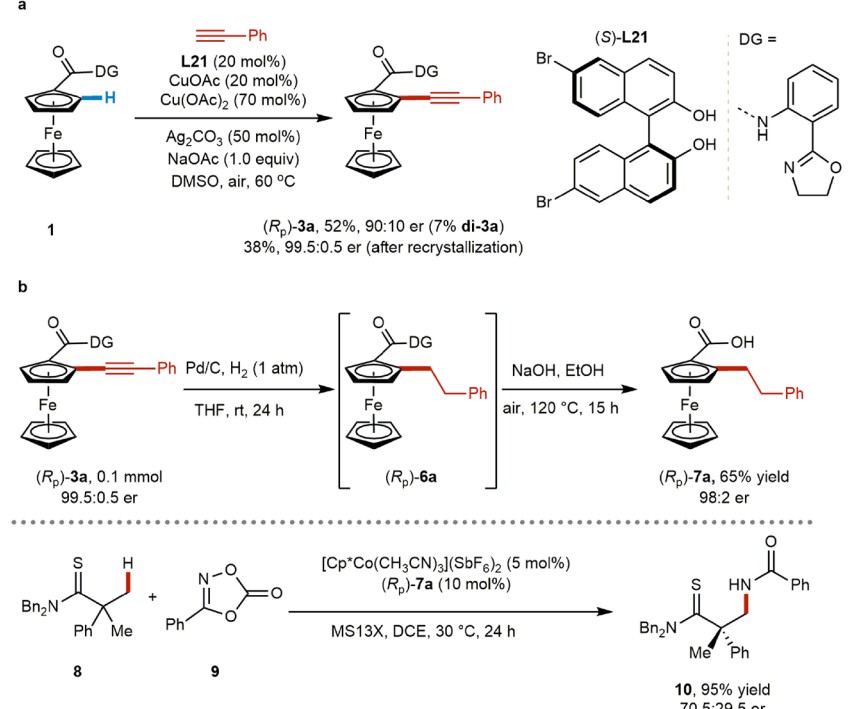

**Fig. 7 | Gram-scale reaction, product elaboration, and synthetic application. a** Gram-scale reaction. **b** Product elaboration and synthetic application in catalysis.

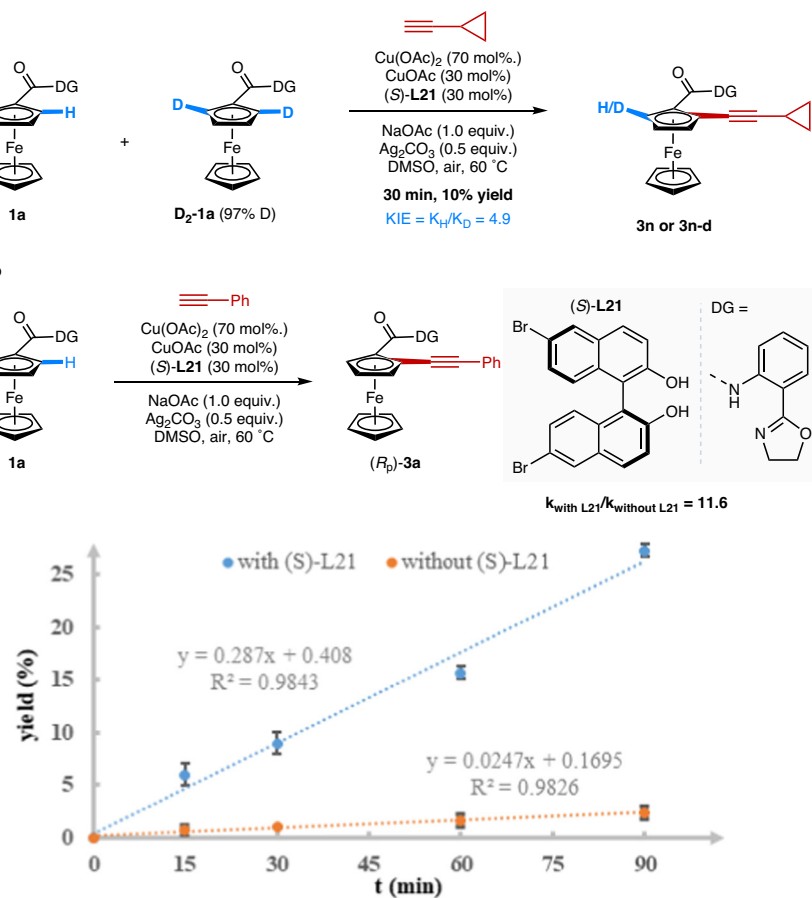

**Fig. 8 | Mechanistic studies. a** Kinetic isotope effect experiments. **b** Initial reaction rate in comparison with and without (*S*)-**L21**.

supporting our initial ligand acceleration hypothesis. Taken together, these experiments indicate that the BINOL scaffold accelerates the key rate- and enantio-determining C–H cleavage step at Cu, similar to the behavior of MPAAs and related ligands for Pd[5–7].

In conclusion, Cu-mediated enantioselective *ortho* C–H alkynylation of ferrocene carboxylic acid derivatives has been realized using a chiral BINOL-derived diol ligand. This scaffold demonstrates ligand-accelerated catalysis with the Cu-mediated C–H activation, which both boosts the reactivity and also enables enantioselective catalysis. The development of additional Cu-catalyzed/mediated asymmetric C–H activation reactions with these chiral diol ligands is an ongoing research direction in our laboratory.

## Methods

### General procedure for Cu-mediated asymmetric C–H alkynylation of ferrocenes

A 15 mL scale tube was charged with substrate **1** or **4** (0.1 mmol, 1.0 equiv.), Cu(OAc)$_2$ (12.7 mg, 0.07 mmol), CuOAc (3.7 mg, 0.03 mmol), (*S*)-**L21** (13.3 mg, 30 mol%), Ag$_2$CO$_3$ (13.8 mg, 0.05 mmol), NaOAc (8.2 mg, 0.1 mmol), **2** (0.25 mmol, 2.5 equiv.), DMSO (5.0 mL) under air atmosphere. The tube was capped tightly, and the reaction mixture was stirred at room temperature for 30 s and then stirred at 60 °C for another 12 h. Upon completion, the reaction was cooled to room temperature, and then EtOAc was added to dilute the reaction mixture. The organic layer was washed with NH$_3$·H$_2$O, saturated brine, and dried over Na$_2$SO$_4$. Volatiles were removed under a vacuum. The crude product was purified by column chromatography to afford the desired product (*R*$_p$)-**3** or (*R*$_p$)-**5**. The ratio of mono/di was determined by the analysis of crude $^1$H NMR. Full experimental details and characterization of new compounds can be found in the Supplementary Methods.

## Data availability

X-ray structural data of compound (*R*$_p$)-**3a** (ccdc 2192815) is available free of charge from the Cambridge Crystallographic Data Center via www.ccdc.cam.ac.uk/data_request/cif. Other chemical characterizations and kinetics data are provided in the Supplementary Information file. All other data are provided in the Supplementary Information file or from the corresponding author upon request.

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

## Acknowledgements

We gratefully acknowledge the National Key R&D Program of China (2021YFA1500200), the National Natural Science Foundation of China (22171277, 22101291, 21821002), the Program of Shanghai Academic Research Leader (23XD1424500), Shanghai Institute of Organic Chemistry, and State Key Laboratory of Organometallic Chemistry for financial support. We also thank H.-C. Shen for verifying the reproducibility of this work.

## Author contributions

X.K. performed the experiments and analyzed the data. J.-J.L. and L.T. assisted with some substrate preparation. K.W. helped prepare the manuscript. P.W. directed the project. C.-H.D., P.W., and J.-Q.Y. conceived the concept and prepared this manuscript with feedback from X.K.

## Competing interests

The authors declare no competing interests.
