## [Peer Review File · Nature Communications]

REVIEWER COMMENTS

Reviewer #1 (Remarks to the Author):

Yu, Wang et al achieved Cu-mediated enantioselective C-H alkynylation of ferrocenes with BINOLs. Ligand acceleration was observed with BINOL derivatives and the authors applied to enantioselective C-H activation of ferrocenes.

The strong point of this manuscript is that the authors realized enantioselective Cu-mediated C-H activation with catalytic amount of chiral source, 6,6'-Br-BINOL, for the first time.

On the other hand, the weak point of this method is the catalytic activity of the protocol, especially in the enantioselective variant. In Fig 6, the combined yield of mono- and di-adducts are presented but the desired chiral mono-adduct was obtained in only moderate yield. Overreaction may be the reason, but I am afraid that the overreaction was required to achieve high enantioselectivity possibly via selective di-functionalization of the minor enantiomer of mono adducts. The yield of mono-adducts in Fig 6 was <50% in 10 entries out of 14 entries, which unfortunately limited the synthetic advantage of the method. The high catalytic loading, 30 mol % of chiral ligand and stoichiometric amount of Cu source, is still acceptable as this is the first example. But, the low yield of desired mono-adducts restricted the synthetic utility of the method. Compared with Pd, Rh, Ir (ref 5-24) as well as with other base metal catalysis (ref 25-32), synthetic utility of Cu-system is not sufficient for publication in Nature Communications.

As to the mechanistic studies, the authors emphasized that the Cu/BINOL system activated C-H bonds in a similar manner as Pd(II). The evidence proposed, however, was not sufficient. KIE and ligand acceleration effects were observed. So, the authors hypothesis is reasonable, but is not fully convinced. 1) The mechanistic studies should be performed with chiral system, using 6,6-Br-BINOL L21. 2) Further support like DFT calculation is required to exclude the SET pathway and clarify the role of BINOLs in C-H activation process.

Overall, this is the first example of Cu-mediated asymmetric C-H activation with BINOL ligands. But, the synthetic utility of the protocol is not sufficient and the evidence for the proposed mechanism is also limited. In its current stage, the manuscript does not reach to the level required for publication in Nature Communications. The manuscript may become suitable for the journal, but significant revisions are required.

- a) Fig 6. Yield of mono-adducts should be presented to clearly show the data for readers.
- b) Is it possible to further optimize the reaction conditions in Fig 6 to maximize the amount of mono-adduct, while keeping good er?
- c) Mechanistic studies in Fig 8 should be performed under enantioselective conditions as the enantio-induction is the key achievement of this article.
- d) In Fig 2, the authors used chiral L-15 and compared its reactivity with racemic BINOL derivatives. I think it is necessary to compare the reactivity using either racemic ligands or chiral ligands, because the reactivity of racemic and chiral ligands may be different like many precedents. Chiral L10, L13 and L14 should be used in Fig 2. Alternatively, the authors can use racemic L15, but chiral L15 was also used in Fig 3 and Fig 4, and so it would be easier to change L10, L13 and L14 into chiral ones.
- e) Additional mechanistic study to support proposed mechanism is required. For example, DFT calculation to clarify the role of BINOL ligands will be highly useful. As the finding of acceleration with BINOLs is the key of this article, more evidence on this aspect is desirable.

Reviewer #2 (Remarks to the Author):

The paper by Wang, Yu, and coworkers describes Cu-mediated enantioselective C-H alkynylation of directing group-containing ferrocenes with BINOL ligands. The paper is well-written and has some novelty due to enantioselective C-H functionalization under copper catalysis; thus, it can be published in Nature Communications despite relatively low general interest in reaction products. The following minor revisions need to be done before the paper is published:

- (1) Authors say in introduction and paper that Cu(II) cleaves C-H bonds. Can bonds be cleaved at Cu(III) oxidation state, and what proof there exists that in this system C-H activation step occurs at Cu(II)?
- (2) The reactions are stoichiometric in Ag. This in some ways destroys the purpose of using base metal Cu for catalysis since a lot of precious Ag is used. It would be nice if in future precious metal reoxidants would be avoided for first-row metal catalysis. This is more of a general comment, rather than what needs to be fixed in the paper.
- (3) For enantioselective reactions, authors usually run reactions to relatively high conversions to dialkynylated product. This inflates enantioselectivity, as minor monoalkynylation enantiomer is likely more reactive in forming disubstitution product than the major one. Please run a selection of reaction to a very low conversion (when di-substituted product is not formed) to determine the inherent enantioselectivity of reaction.

(4) In SI, please report R_f values for compounds that were chromatographed. Please report ¹³C to one decimal.

Reviewer #3 (Remarks to the Author):

Yu, Wang, and coworkers describe an impressive and novel protocol for the synthesis of chiral ferrocenes through Cu-mediated C-H alkynylation. BINOL-derived ligands were found to accelerate the reaction effectively in an asymmetric manner, giving rise to the products with good enantioselectivities, which is so far the first example of asymmetric C-H activation by Cu. The current work also represents an elegant example of the ligand acceleration effect in the CMD process of Cu in addition to commonly used noble metals such as Pd, Rh, Ir, etc. Therefore, I believe this work could definitely gain broad interest and be beneficial to the field of asymmetric C-H functionalization. Overall, I am very supportive of this work in nature communication.

The authors should address the following minor issues:

1. In the part of asymmetric alkynylation, the authors used a combined Cu source CuOAc and Cu(OAc)₂. What is the role of this combination?
2. A related Ir-catalyzed enantioselective C-H activation of ferrocenes should be cited, ACS catalysis 2022, 12, 1830-1840.

Reviewer 1 thought this manuscript might become suitable after significant revisions.

1. Comment: “Fig 6. Yield of mono-adducts should be presented to clearly show the data for readers.”

Response: We thank the reviewer’s advice. We now reported the yields mono-alkynylated product in Fig 6.

2. Comment: “Is it possible to further optimize the reaction conditions in Fig 6 to maximize the amount of mono-adduct, while keeping good er?”

Response: We thank the reviewer for this comment. We have systematically evaluated the reaction parameters, including ligands, temperature, catalyst loading (see section 2.3 in the Supplementary Information) to improve the mono/di selectivity in our asymmetric reaction. Unfortunately, it is difficult to simultaneously achieve high mono/di selectivity and high enantioselectivity when optimizing reaction parameters.

3. Comment: “Mechanistic studies in Fig 8 should be performed under enantioselective conditions as the enantio-induction is the key achievement of this article”

Response: We thank the reviewer’s input. Following this recommendation, we have conducted the KIE experiments and kinetic profiles under our optimized enantioselective conditions using ligand (*S*)-**L21**. The results of the KIE experiments and kinetic studies with (*S*)-**L21** were consistent with the previous results with **L15**. We have replaced Fig. 8 with the new results in the manuscript, and have incorporated the results with **L15** in the Supplementary Information.

(*S*)-**L21** KIE = 4.9, $k_{L21}/k_{noL} = 11.6$. **L15** (under racemic conditions) KIE = 2.8, $k_{L15}/k_{noL} = 6.2$.

4. Comment: “In Fig 2, the authors used chiral L-15 and compared its reactivity with racemic BINOL derivatives. I think it is necessary to compare the reactivity using either racemic ligands or chiral ligands, because the reactivity of racemic and chiral ligands may be different like many precedents. Chiral L10, L13 and L14 should be used in Fig 2. Alternatively, the authors can use racemic L15, but chiral L15 was also used in Fig 3 and Fig 4, and so it would be easier to change L10, L13 and L14 into chiral ones.”

Response: We thank the reviewer for this comment. To address this concern, we have conducted the reactions with (*S*)-L10, (*S*)-L13, (*S*)-L14 under our racemic conditions, and revised the Fig. 2 in the manuscript. As showed below, the results with enantioenriched ligands (*S*)-L10, (*S*)-L13 and (*S*)-L14 were the same as with the racemic mixtures. We have added these results to the manuscript.

previous results with racemic ligands:

(±)-L10, 68%
mono/di = 22/1.0

(±)-L13, 62%
mono/di = 30/1.0

(±)-L14, 47%
mono/di > 25/1.0

results with chiral ligands:

(S)-L10, 69%
mono/di = 22/1.0

(S)-L13, 62%
mono/di = 30/1.0

(S)-L14, 48%
mono/di > 25/1.0

5. Comment: “Additional mechanistic study to support proposed mechanism is required. For example, DFT calculation to clarify the role of BINOL ligands will be highly useful. As the finding of acceleration with BINOLs is the key of this article, more evidence on this aspect is desirable.”

Response: We thank the reviewer’s advice. In order to understand the role of BINOL ligands in the catalytic cycle, specifically their involvement and acceleration C–H cleavage step, extensive kinetic studies and characterization of catalytic intermediates, particularly with regards to C–H cleavage, are first necessary to identify the speciation and structural composition of the catalytic species performing C–H cleavage. Without this crucial information, computational studies alone will not add much mechanistic understanding of the key elementary step nor the catalytic cycle. We believe that in-depth mechanistic studies, given their complexity and detail, are best suited for a separate manuscript. Our current report is intended to provide immediate disclosure of the BINOL ligand acceleration of Cu-catalyzed C–H activation, a phenomena and method of significant interest to the catalysis community.

Reviewer 2 recommends the publication of this manuscript with minor revision.

1. Comment: “Authors say in introduction and paper that Cu(II) cleaves C-H bonds. Can bonds be cleaved at Cu(III) oxidation state, and what proof there exists that in this system C-H activation step occurs at Cu(II)?”

Response: We thank the reviewer for this comment. Although we don't have direct experimental support for Cu(II) cleaving C–H bonds in this specific system, our previous mechanistic studies of Cu-catalyzed C–H activation suggests Cu(II) C–H cleavage via a CMD mechanism (*ACS Catal.* **2021**, *11*, 12620). Detailed mechanistic and computational studies are ongoing and will be reported in due course.

2. Comment: “The reactions are stoichiometric in Ag. This in some ways destroys the purpose of using base metal Cu for catalysis since a lot of precious Ag is used. It would be nice if in future precious metal reoxidants would be avoided for first-row metal catalysis. This is more of a general comment, rather than what needs to be fixed in the paper.”

Response: We agree with the referee's comments. The use of stoichiometric Ag salts represents a practical drawback. We are working on this issue and will disclose silver free Cu-catalyzed C–H activation reactions in due course.

3. Comment: “For enantioselective reactions, authors usually run reactions to relatively high conversions to dialkynylated product. This inflates enantioselectivity, as minor monoalkynylation enantiomer is likely more reactive in forming disubstitution product than the major one. Please run a selection of reaction to a very low conversion (when di-substituted product is not formed) to determine the inherent enantioselectivity of reaction.”

Response: We appreciate this thoughtful input from the reviewer. We have data which shows that kinetic resolution of the monoalkynylated product, as alluded to by the reviewer, is unlikely. When the model reaction is run to low conversion, 8% yield (by NMR) of the monoalkynylated product (no di product) is observed in 89.5:10.5 er. Compared with the results after 12 hours (39% mono/24% di and 92.5:7.5 er), there is only a slight increase in enantioselectivity, indicating that chiral amplification or kinetic resolution of the mono-product is unlikely.

4. Comment: “In SI, please report Rf values for compounds that were chromatographed. Please report ¹³C to one decimal.”

Response: We thank the reviewer for this comment. To address this concern, we have added the Rf values for all compounds, and revised the ¹³C NMR data to one decimal in the SI.

Reviewer 3 recommends the publication of this manuscript with minor revision (very supportive).

1. Comment: “In the part of asymmetric alkylation, the authors used a combined Cu source CuOAc and Cu(OAc)₂. What is the role of this combination?”

Response: We thank the reviewer’s comment. Control experiments have been conducted to elucidate the role of CuOAc and Cu(OAc)₂. As shown below, Cu(OAc)₂ is essential for achieving high enantioselectivity. Only 65:35 er and 24% yield were obtained in the absence of Cu(OAc)₂, whereas when only Cu(OAc)₂ is used, high (90:10 er) enantioselectivity is still observed but with lower yield (49%). Although the precise roles of the two Cu salts are unclear, these experiments indicate that Cu(OAc)₂ is likely the active catalyst for the enantioselective C–H activation event.

Entry	x (mol%)	y (mol%)	Yield (%)	mono/di	e.r.	1a (%)
1	70	30	64	1.6/1.0	92.5:7.5	2
2	-	30	24	7.0/1.0	65:35	65
3	70	-	49	3.9/1.0	90:10	28

2. Comment: “A related Ir-catalyzed enantioselective C-H activation of ferrocenes should be cited, ACS catalysis 2022, 12, 1830-1840.”

Response: We thank the reviewer for pointing out this omission. We have cited this reference as Ref 20.

REVIEWERS' COMMENTS

Reviewer #1 (Remarks to the Author):

The authors revised the manuscript in responses to the requests from all reviewers. Although DFT calculation was not added, other revisions were reasonably performed and additional data improved the quality of the manuscript. Now, I recommend publication of this manuscript as it stands.

Reviewer #2 (Remarks to the Author):

OK to publish, authors have fixed the issues.